# OpenReview forum: "Safe RLHF: Safe Reinforcement Learning from Human Feedback"
_ICLR.cc/2024/Conference — ICLR 2024 spotlight_

### Official Review · Reviewer_NMqi · 2023-10-24

**Soundness:** 2 fair
**Presentation:** 2 fair
**Contribution:** 2 fair
**Rating:** 8
**Confidence:** 3

**Summary:**

The paper introduces a novel algorithm, Safe Reinforcement Learning from Human Feedback (Safe RLHF), to address the crucial challenge of balancing the performance and safety of large language models (LLMs). LLMs often face an inherent tension between the objectives of being helpful and harmless, which can confuse crowdworkers during training. Safe RLHF effectively decouples human preferences related to helpfulness and harmlessness, enabling separate reward and cost models. The safety concern of LLMs is formalized as an optimization task to maximize the reward function while satisfying specified cost constraints. Using the Lagrangian method, Safe RLHF dynamically adjusts the balance between these two objectives during fine-tuning. Experimental results demonstrate that three rounds of fine-tuning using Safe RLHF significantly improve the helpfulness and harmlessness of LLMs, surpassing existing value-aligned algorithms. This work is a significant contribution to enhancing the safety of AI systems based on LLMs by effectively addressing the tension between helpfulness and harmlessness during fine-tuning, offering a promising approach to mitigate harmful responses while improving model performance.

**Strengths:**

The paper's strength lies in its innovative approach, Safe Reinforcement Learning from Human Feedback (Safe RLHF), which addresses the critical challenge of striking a balance between helpfulness and harmlessness objectives in the training of large language models (LLMs). By decoupling human preferences from these objectives, the paper ensures unbiased feedback during data annotation and adaptively balances the trade-off between these inherently conflicting training goals using the Lagrangian method. Notably, Safe RLHF is the first integration of Safe RL and RLHF frameworks, incorporating a two-dimensional human annotation scheme and a safe training mechanism. Through three rounds of Safe RLHF fine-tuning, the paper effectively enhances the helpfulness of the base model while significantly reducing harmful responses, surpassing the performance of existing value-aligned algorithms. The release of all data and training codes enhances the paper's reproducibility and validates its findings, making it a valuable contribution to improving the safety and performance of AI systems based on LLMs.

**Weaknesses:**

- Even though the paper is the first integration of Safe RL and RLHF frameworks, the contribution is limited in the case of Safe RL.
- The comparison between the (reward and cost) model in Section 3.2 to the classic (reward and cost) signals in safe RL should be clarified more.
- The convergence of the proposed methods may be hard to guarantee, and there are no related theoretical results.

**Questions:**

- Figure 1 needs to be polished more, and the font there is a bit small.
- Can we use any Lagrangian safe RL methods (TRPO-Lag, PPO-Lag) in Safe RLHF?
- What is the difference between Safe RLHF and the off-policy Lagrangian safe RL methods, e.g., WCSAC (published in AAAI-21 and MLJ-23)?
- With safe RLHF, how can we ensure the learning is stable and finally converge?
- The stability of the reward and cost signals should be analyzed.

**Details Of Ethics Concerns:**

This paper contains example data that may be offensive or harmful. But there are relevant prompts in the paper.

---

> ### Author Response · Authors · 2023-11-16
> **Official Reply to Reviewer NMqi (1/2)**
>
> > **W1:** Even though the paper is the first integration of Safe RL and RLHF frameworks, the contribution is limited in the case of Safe RL.
>
> **Reply to W1:** Our primary goal is to propose **a new RLHF framework that addresses the challenging tension** between helpfulness and harmlessness goals in the field of LLM alignment, rather than introducing a new Safe RL algorithm. Instead, we aim to broaden the application scope of Safe RL algorithms, bringing new perspectives to both fields.
>
> Here, we reiterate our contributions:
>
> - Safe RLHF is **the first work to combine Safe RL with the RLHF framework**.  It is a comprehensive methodology that includes decoupled data collection, training of two different preference models, and fine-tuning through the integration of Safe RL. It formalizes safety as a constraint and dynamically adjusts it during training, aiming to navigate the inherent tension between helpfulness and harmlessness.
> - Our **extensive experiments** demonstrate the following conclusions:
>
>     - Fine-tuning with Safe RLHF framework effectively enhances the helpfulness and harmlessness of LLMs under human values (Section 5.2.1).
>     - Decoupling helpfulness from harmlessness effectively increases agreement among crowd workers when annotating preference data, reducing bias due to the contradiction (Section 5.2.2).
>     - Compared to a fixed ratio of optimizing for helpfulness and harmlessness, dynamically adjusting the trade-off between them during training more effectively guides the inherent tension between the two (Section 5.2.3).
>     - Providing preference-based scores, rather than just a binary safe/unsafe classifier, improves LLM safety more efficiently (Section 5.2.4).
>
> - We **release all the data and code** from our experiments in hopes of **contributing to the reduction of the high research costs in this field**. (Currently, they are in the supplementary materials.)
>
> **We sincerely hope that you could appreciate our paper and recognize our efforts and contributions.**
>
> ---
>
> > **W2:** The comparison between the (reward and cost) model in Section 3.2 to the classic (reward and cost) signals in safe RL should be clarified more.
>
> **Reply to W2:** Compared to traditional Safe RL's reward and cost signals, Safe RLHF presents the following challenges:
>
> - **Human values are high-dimensional and complex, making it impossible to design reward and cost functions in a rule-based manner.** Instead, human preferences must be modeled from collected datasets. Thus, guiding crowd workers to provide high-quality annotations (Section 4.1), constructing prompt datasets for large language models (Section 5.1 Prompts and Red-teaming), controlling the distribution of prompt-response pairs in the dataset (Section 5.1 Preference Datasets), and training better preference models (Section 4.2) are among the issues that a comprehensive large language model alignment process must address.
>
> - The threshold for what is considered safe in human values cannot be formulaically defined. Therefore, **we propose a new training method for a Cost Model to model the safety threshold**, which is one of our contributions. Building on the Bradley-Terry model, we introduce a comparison loss between responses and a virtual response $y_0$ near the safety threshold, setting the cost of $y_0$ to zero. This design separates safe from unsafe responses at a cost boundary of zero (as shown in Figure 2(a)), making zero the safety threshold for an individual response.

---

> > ### Author Response · Authors · 2023-11-16
> > **Official Reply to Reviewer NMqi (2/2)**
> >
> > > **W3:** The convergence of the proposed methods may be hard to guarantee, and there are no related theoretical results.
> >
> > > **Q4:** With safe RLHF, how can we ensure the learning is stable and finally converge?
> >
> > **Reply to W3Q4**: First, we must acknowledge that **due to the high-dimensional complexity of LLMs and the randomness** in training preference models, **it is difficult to theoretically guarantee the convergence of RLHF**. This is a common scenario in this research field. The theoretical interpretability of LLMs has become an independent and challenging area of research, and it is difficult for us to pursue parallel studies in this work. Therefore, I sincerely hope that the reviewers will understand the difficulty of this matter in the field of LLMs and forgive our lack of theoretical analysis on convergence.
> >
> > Second, **In the RLHF framework, training often needs to be early stop before convergence** due to the issue of **over-optimization [1]**. The reward model and cost model are learned from human preference data. **They are proxies of human preference.** They can accurately predict the scores only within a finite range. Continuously training on the same reward model and cost model can easily lead to a phenomenon known as **'reward hacking' [2]**. Thus, an early stop before over-optimization is necessary. In our experiments, we also observed that when LLMs converge with the current reward model and cost model, it usually indicates the occurrence of reward hacking.
> >
> > Third, **empirically**, through three iterations of Safe RLHF, we have validated that this framework is stable and efficient to enhance both the helpfulness and harmlessness of large language models simultaneously (Section 5.2.1). In our experiments, **the primary measure for stability is to conduct periodic evaluations with human evaluators and early stop training** in cases of over-optimization.
> >
> > **References:**
> >
> > [1] Gao, Leo, John Schulman, and Jacob Hilton. "Scaling laws for reward model overoptimization." International Conference on Machine Learning. PMLR, 2023.
> >
> > [2] Di Langosco, L. L., Koch, J., Sharkey, L. D., Pfau, J., & Krueger, D. Goal misgeneralization in deep reinforcement learning. In International Conference on Machine Learning (pp. 12004-12019). PMLR.
> >
> > ---
> >
> > > **Q1:** Figure 1 needs to be polished more, and the font there is a bit small.
> >
> > **Reply to Q1:** We are very grateful to the reviewer for their valuable feedback on Figure 1. We will actively incorporate these improvements in the newly submitted version of our paper.
> >
> > ---
> >
> > > **Q2:** Can we use any Lagrangian safe RL methods (TRPO-Lag, PPO-Lag) in Safe RLHF?
> >
> > **Reply to Q2:** We conservatively believe that **first-order algorithms** in the Safe RL field, such as PPO-Lag, FOCOPS, P3O, etc., **hold promise for application within the Safe RLHF framework**.
> >
> > However, for **second-order algorithms** like TRPO-Lag, CPO, etc., which require solving the inverse of the Fisher matrix through conjugate gradient methods, **the computational resources needed for LLMs might be incredibly vast**.
> >
> > ---
> >
> > > **Q3:** What is the difference between Safe RLHF and the off-policy Lagrangian safe RL methods, e.g., WCSAC (published in AAAI-21 and MLJ-23)?
> >
> > **Reply to Q3:**  The RL phase of Safe RLHF should be classified as an on-policy algorithm, where the model used for updates and inference is the same. We carefully read WCSAC, as recommended by the reviewer, and find that **combining off-policy Safe RL may offer some distinct advantages**. For instance, it might reduce the communication bottlenecks during the inference phase of LLM training, and the maximum entropy framework could enhance sampling diversity.
> > We are very grateful to the reviewer for providing valuable paper references and research discussions. **We will include this part in the 'Future Work' section of the revised paper.**
> >
> > ---
> >
> > > **Q5:** The stability of the reward and cost signals should be analyzed.
> >
> > **Reply to Q5:** The reward model and cost model are trained from human preference datasets. Listed below are factors related to their stability, along with the corresponding measures taken:
> >
> > - **Dataset Quality:** As mentioned in W2A, we implemented additional optimizations in the annotation process and data distribution.
> >
> > - **Randomness in Deep Learning:** We evaluate their predictive accuracy on a reserved test set and select the model with the highest accuracy (>75%) for the next step.
> >
> > - **Over-optimization leading to preference model failure:** As described in Reply to W3Q4, we periodically test and stop prematurely before over-optimization occurs.

---

> > > ### Comment · Reviewer_NMqi · 2023-11-18
> > >
> > > Thanks to the authors for their response.
> > >
> > > In general, my concerns were addressed. I am happy to raise my score.

---

> > > > ### Author Response · Authors · 2023-11-19
> > > > **Thanks Reviewer NMqi for Approving Our Work**
> > > >
> > > > We sincerely appreciate your acknowledgment and are deeply encouraged by your decision to raise our rating. It is our honor to address your concerns, which have been helpful to our work and will be integral to the improvements in our final version.

---

### Official Review · Reviewer_xNog · 2023-10-31

**Soundness:** 3 good
**Presentation:** 3 good
**Contribution:** 2 fair
**Rating:** 6
**Confidence:** 5

**Summary:**

The paper introduces Safe Reinforcement Learning from Human Feedback (Safe RLHF), a novel algorithm designed to address the challenge of balancing helpfulness and harmlessness in Large Language Models (LLMs). Safe RLHF decouples human preferences regarding helpfulness and harmlessness, allowing for separate training of reward and cost models. By leveraging the Lagrangian method, Safe RLHF dynamically adjusts the balance between these objectives during fine-tuning. Experimental results demonstrate significant improvements in both helpfulness and harmlessness compared to existing value-aligned algorithms.

**Strengths:**

1. This paper is well-written and easy to follow. The authors present a well-defined methodology, including a clear description of the Safe RLHF pipeline, preference annotation process, and training algorithms for reward and cost models.

2. Given the societal impact of LLMs, ensuring their safety and usefulness is of utmost importance. Safe RLHF presents a significant contribution by effectively aligning human values with model behavior, addressing an essential concern in AI research.

**Weaknesses:**

1. The technique contributions seem incremental to me. The decoupling of rewards into rewards and costs is a standard formulation in CMDP, and the Lagrangian methods with RL are not new at all.

2. Another concern in this paper is that I don't think there is an appropriate cost threshold and cost-reward trade-off in the LLM alignment settings. I think safety is cleary a priority when compared with preferences. So that being said, how do you define how much safety LLMs are to trade off the preference performance? If this is a super safe-critical scenario, the cost limit should be 0. That being said. A simple LLM finetuning reward function can be designed that A>B if (preference (A>B) and (A passes safety check threshold)). I do not see the necessity of using the cost-reward formulation of CMDP.

3. Also, this concern is a follow-up to point 2. I think the authors failed to convince me that standard rewarding shaping is not good enough in this setting. Since Lagrangian methods need to try different threshold values, It is not enough to conclude Lagrangian methods is better than reward shaping for LLMs fine-tuning without trying different reward shaping coefficients.

**Questions:**

1. Could you elaborate on the limitations and potential risks associated with Safe RLHF? For instance, are there scenarios where the algorithm might fail to balance helpfulness and harmlessness effectively? Understanding the limitations would provide a more nuanced perspective on the applicability of your approach.

2. The paper lacks a discussion on the computational resources and time required for the Safe RLHF training process. Can you provide insights into the computational efficiency and scalability of your method, especially concerning large-scale LLMs?

3. As for safe RL methods like Lagrangian methods, the cost threshold is super sensitive. A threshold that is too small might result in complete failures of policy training. I noticed that the authors selected negative cost thresholds for two experiments. How many values of cost threshold have you tried? And how robust is the Lagrangian method for RLHF to different threshold values?

---

> ### Author Response · Authors · 2023-11-16
> **Official Reply to Reviewer xNog (1/3)**
>
> > **W1:** The technique contributions seem incremental to me. The decoupling of rewards into rewards and costs is a standard formulation in CMDP, and the Lagrangian methods with RL are not new at all.
>
> **Reply to W1:** We sincerely hope that reviewers will recognize that **our work is not incremental**, especially in **the domain of LLM safety**.
>
> Logically, we firstly decouple human value towards LLMs into two parts: helpfulness and harmlessness. After modeling harmlessness as a constraint that needs to be prioritized, we formalize the entire problem under the CMDP framework. Reversing this logical order is a disheartening accusation. It overshadows our efforts in
>
> - **guiding the crowdworkers to generate decoupling data** reflecting their preferences for helpfulness and harmlessness;
> - utilizing reward model and **our proposed novel cost model** to represent objective and constraint signals;
> - **integrating Safe RL with the training of LLMs.**
>
> Standard CMDPs indeed encompass both reward and cost signals. However, our innovation lies in how to **model, acquire, and utilize** such signals in the alignment of LLMs.
>
> Furthermore, we do not deny that the Lagrangian method is a mature technology in the field of traditional Safe RL. However, for our comprehensive Safe RLHF framework, it constitutes just one component. We do not claim the introduction of the Lagrangian method to solve Safe RL problem as our contribution. Here, we reiterate our contributions:
>
> - Safe RLHF is **the first work to combine Safe RL with the RLHF framework**.  It is a comprehensive methodology that includes decoupled data collection, training of two different preference models, and fine-tuning through the integration of Safe RL. It formalizes safety as a constraint and dynamically adjusts it during training, aiming to navigate the inherent tension between helpfulness and harmlessness.
> - Our **extensive experiments** demonstrate the following conclusions:
>
>     - Fine-tuning with Safe RLHF framework effectively enhances the helpfulness and harmlessness of LLMs under human values (Section 5.2.1).
>     - Decoupling helpfulness from harmlessness effectively increases agreement among crowd workers when annotating preference data, reducing bias due to the contradiction (Section 5.2.2).
>     - Compared to a fixed ratio of optimizing for helpfulness and harmlessness, dynamically adjusting the trade-off between them during training more effectively guides the inherent tension between the two (Section 5.2.3).
>     - Providing preference-based scores, rather than just a binary safe/unsafe classifier, improves LLM safety more efficiently (Section 5.2.4).
>
> - We **release all the data and code** from our experiments in hopes of **contributing to the reduction of the high research costs in this field**. (Currently, they are in the supplementary materials.)

---

> > ### Author Response · Authors · 2023-11-16
> > **Official Reply to Reviewer xNog (2/3)**
> >
> > > **W2:** Another concern in this paper is that I don't think there is an appropriate cost threshold and cost-reward trade-off in the LLM alignment settings. I think safety is cleary a priority when compared with preferences. So that being said, how do you define how much safety LLMs are to trade off the preference performance? If this is a super safe-critical scenario, the cost limit should be 0. That being said. A simple LLM finetuning reward function can be designed that A>B if (preference (A>B) and (A passes safety check threshold)). I do not see the necessity of using the cost-reward formulation of CMDP.
> >
> > **Reply to W2:** Firstly, **many prior work [1,2,3,4] has noted that the trade-off between helpfulness and harmlessness during training remains a critical challenge in the field of LLM alignment.** This is due to the inherent tension between these two optimization goals and the complex, high-dimensional nature of human values that are hard to represent with rule-based approaches. Therefore, this is not a task that can be overlooked or simplified.
> >
> > Secondly, we also believe that harmlessness should be prioritized over helpfulness. As stated in our paper, the trade-off that needs to be achieved during training is: **to maximize helpfulness as much as possible, while ensuring that the output aligns with human values of safety.** Therefore, we model safety as a constraint that must be met first. Naturally, compared to the traditional RLHF's MDP framework, it becomes necessary for us to formalize this problem within the CMDP framework.
> >
> > Thirdly, **using a simple reward format to express both helpfulness and harmlessness is the approach taken by traditional RLHF.** As many previous works [1,2,3,4] have mentioned, such approach poses significant challenges during data annotation and training due to the tension between helpfulness and harmlessness. Meanwhile, in section 5.2.2, we empirically demonstrated the shortcomings of the traditional single-reward RLHF method. Compared to Safe RLHF, it results in lower data agreement, as well as reduced efficiency in optimizing both helpfulness and harmlessness, as illustrated in Figure 6(a).
> >
> > **References:**
> >
> > [1] Kenton, Zachary, et al. "Alignment of language agents."
> >
> > [2] Bai, Yuntao, et al. "Training a helpful and harmless assistant with reinforcement learning from human feedback."
> >
> > [3] Glaese, Amelia, et al. "Improving alignment of dialogue agents via targeted human judgements."
> >
> > [4] Zhao, Wayne Xin, et al. "A survey of large language models."
> >
> > ---
> >
> > > **W3:** Also, this concern is a follow-up to point 2. I think the authors failed to convince me that standard rewarding shaping is not good enough in this setting. Since Lagrangian methods need to try different threshold values, It is not enough to conclude Lagrangian methods is better than reward shaping for LLMs fine-tuning without trying different reward shaping coefficients.
> >
> > **Reply to W3:** It is important to note that **there seems to be a misunderstanding** among the reviewers regarding the comparative experiments between Safe RLHF and Reward Shaping methods. Our experiments extensively tested **seven different reward shaping coefficients**, ν, specifically at values of **0.01, 0.5, 1, 2, 5, 10, and 100**, as depicted in **Figure 6b**. As mentioned in the paper, these seven sets of coefficients almost cover the coefficient space. Extremely high (ν = 5, 10, 100) and extremely low (ν = 0.01, 0.5) Reward Shaping weights lead to the over-optimization of one objective at the detriment of the other. Moderate reward shaping weights (ν = 1, 2) are still ineffective in resolving the conflict between the objectives of helpfulness and harmlessness, with their enhancements remaining subpar compared to Safe RLHF. Therefore, we believe that this number of variations is sufficient to demonstrate the superiority of our method over the Reward Shaping approach.
> >
> > > **Q1:** Could you elaborate on the limitations and potential risks associated with Safe RLHF? For instance, are there scenarios where the algorithm might fail to balance helpfulness and harmlessness effectively? Understanding the limitations would provide a more nuanced perspective on the applicability of your approach.
> >
> > **Reply to Q1:** In fact, **our paper already includes an analysis of its limitations**. Due to space constraints, this analysis is located in **Appendix A**. In brief, in our experiments, we lacked high-quality SFT data for SFT and PTX loss; we did not specifically optimize for multi-turn dialogue scenarios; and as our algorithm is a type of deep learning algorithm, it requires collaboration with additional mechanisms (such as pre- and post-check strategies) in real-world applications to achieve a safer AI system.

---

> > > ### Author Response · Authors · 2023-11-16
> > > **Official Reply to Reviewer xNog (3/3)**
> > >
> > > > **Q2:** The paper lacks a discussion on the computational resources and time required for the Safe RLHF training process. Can you provide insights into the computational efficiency and scalability of your method, especially concerning large-scale LLMs?
> > >
> > > **Reply to Q2:** All experiments in this paper utilized a large language model with 7 billion parameters.
> > > The server's CPU was an Intel(R) Xeon(R) Platinum 8378A CPU @ 3.00GHz with 64 cores, and the graphics cards were NVIDIA A800-SXM4-80GB $\times 8$, with NVLink support and the graphics driver version being 525.125.06.
> > >
> > > The average time required for a single round of data collection in Safe RLHF was approximately **two weeks for crowdworker annotation** and **one week for professional quality control**.
> > > Training for a single round of Safe RLHF required between **10 to 20 hours**, with the specific time dependent on the average length of inference. The total cost for the related data annotations was around **70,000 U.S. dollars**, and the total cost for the related training equipment was about **120,000 U.S. dollars**.
> > >
> > > We **release all the data and code** from our experiments in hopes of contributing to the reduction of the high research costs in this field.
> > > This discussion is relevant for other researchers attempting to replicate our work, therefore we will include it in the appendix of our paper.
> > >
> > > ---
> > >
> > > > **Q3:** As for safe RL methods like Lagrangian methods, the cost threshold is super sensitive. A threshold that is too small might result in complete failures of policy training. I noticed that the authors selected negative cost thresholds for two experiments. How many values of cost threshold have you tried? And how robust is the Lagrangian method for RLHF to different threshold values?
> > >
> > > **Reply to Q3A:**
> > > Firstly, we introduce a new training methodology for the Cost Model, specifically tailored to establish an appropriate threshold.  **Our Cost Model allows us to use a cost of 0 as the threshold for determining whether a response to a prompt is safe.** Compared to conventional preference modeling methods, our Cost Model significantly improves the robustness of the cost threshold in Safe RLHF. To illustrate this point, during the rebuttal phase, we supplemented our experiments with a traditional ordinal model used as a Cost Model, as shown in the figure:
> > >
> > > https://anonymous.4open.science/r/Rebuttal-ICLR24/images/cost-model-ablation.png
> > >
> > > As the above Figure shows, using the conventional preference modeling, safe and unsafe responses cannot be distinguished solely based on cost values.
> > >
> > > Secondly, in our experimental setup, **the design of our Cost Model enables us to compute a reference threshold directly**, rather than searching for this hyper-parameter in the real number space, $\mathbb{R}$. The computation process is detailed as follows: we initially utilize the Cost Model to estimate the cost values for the LLM on the training set. Subsequently, we adjust all cost values exceeding zero down to zero and compute their mean. This calculated mean serves as our reference threshold. This threshold essentially reflects the anticipated average cost at the point where responses, currently deemed unsafe, just become safe. This is also why the second and third rounds of Safe RLHF have a negative threshold: most responses of the model on the training set are safe being predicted as negative numbers by the Cost Model.
> > >
> > > In summary, our Cost Model's design enabled us to use a calculated threshold in our experiments, rather than blindly attempting various thresholds. This methodological choice ensures the robustness of the threshold.
> > >
> > > ---
> > >
> > > **Finally, we sincerely hope that the reviewer could re-evaluate our paper and provide a new assessment.**

---

> > > > ### Author Response · Authors · 2023-11-20
> > > > **Hope to Get Your Reply**
> > > >
> > > > Dear Reviewer xNog,
> > > >
> > > > As the deadline is nearing, we wanted to gently follow up on our recent submission. Your feedback is highly valuable to us, and we would appreciate any updates or further guidance you might have regarding our revisions and responses.
> > > >
> > > > Thank you for your time and consideration.

---

> > > > > ### Comment · Reviewer_xNog · 2023-11-20
> > > > > **Appreciation and Further Clarifications on Technical Contributions**
> > > > >
> > > > > I would like to acknowledge the diligent efforts undertaken to address some of my concerns, particularly pertaining to questions Q1, Q2, Q3, and Weakness3. However, I find it necessary to delve deeper into the discussion surrounding the novelty of Safe RLHF and the application of CMDP formulation, specifically addressing Weakness1 and Weakness2.
> > > > >
> > > > > - **The nolvety of Safe RLHF**:
> > > > >
> > > > > Regarding the novelty of Safe RLHF, the authors assert that their primary technical contributions lie in the *decoupling of human value towards LLMs into two distinct components: helpfulness and harmlessness, with a focus on guiding crowdworkers to generate decoupling data.* After revisiting the paper, I remain unconvinced of this being a significant technical contribution. Unlike other works in diverse domains, such as power grid systems [1], where specific challenges and domain requirements are met, I fail to identify a distinctive LLM-specific challenge necessitating innovative safe RL frameworks in this work.
> > > > >
> > > > > The claimed innovation in *modeling, acquiring, and utilizing signals in the alignment of LLMs, specifically through CMDP formulation, guiding crowdworkers, and applying Lagrangian-based methods*, does not seem to present a compelling argument for an ICLR-level paper's technical contributions. While I appreciate the pioneering nature of this work in applying safe RL to LLMs, I believe it may be more suitable for the LLMs track rather than the safe RL track.
> > > > >
> > > > >
> > > > >
> > > > > [1] Glavic, M., Fonteneau, R., & Ernst, D. (2017). Reinforcement learning for electric power system decision and control: Past considerations and perspectives. IFAC-PapersOnLine, 50(1), 6918-6927.
> > > > >
> > > > > - **The necessity of using the cost-reward formulation of CMDP**:
> > > > >
> > > > > On the matter of the necessity of using the cost-reward formulation of CMDP, I appreciate the additional clarifications provided. However, I wish to express further concerns regarding the simplicity of the proposed LLM finetuning reward function. I said in my review that *A simple LLM finetuning reward function can be designed that A>B if (preference (A>B) and (A passes safety check threshold))*. Actually, it might require a lot of efforts to make simple methods works (just like how RLHF worked for LLMs which required a lot of engineering and tuning). However, I understand that the authors could not possibly tune the other baseliens with that much effort. And I agree CMDP formulation is a good way to ease that difficulty.
> > > > >
> > > > >
> > > > > In conclusion, while the authors have made commendable efforts to address some of my concerns, a few lingering reservations persist. I acknowledge the potential value this work holds for the LLMs community, given its pioneering application of safe RL to LLMs alignment, and have adjusted my overall assessment accordingly.
> > > > >
> > > > > Thank you for your attention to these matters, and I look forward to any further clarifications or insights you may provide.

---

> > > > > > ### Author Response · Authors · 2023-11-21
> > > > > > **Appreciation and Further Discussion with Reviewer xNog**
> > > > > >
> > > > > > Thank you for recognizing our efforts, and also for your time and comments. We are honored to have the opportunity to discuss further with you.
> > > > > >
> > > > > > ---
> > > > > >
> > > > > > **Part 1: The Novelty of Safe RLHF**
> > > > > >
> > > > > > We appreciate your acknowledgment of our contributions to the LLM domain. Our main contribution is proposing a safe training framework for LLMs, rather than introducing a new Safe RL algorithm. In response to questions about the technical contributions of our work, we have revisited the technical challenges we address.
> > > > > >
> > > > > > Safe RLHF primarily addresses the following critical challenges in LLMs: **the tension between helpfulness and harmlessness during training**. To tackle this issue, we have integrated the Safe RL framework with LLM training, which also brought several downstream technical challenges which are LLM-specific:
> > > > > >
> > > > > > - There are **no direct Reward and Cost signals in LLM training**, and human values **cannot be given by rule-based functions**. Therefore, we need to collect decoupled human preference data to learn models for Reward and Cost.
> > > > > >
> > > > > > - **The safety threshold in LLMs is an abstract concept.** Traditional preference models are not suitable for fitting safety because there is no numerical threshold under their measurement scale. We need to **learn an abstract concept of safety judgment within a human group** (crowd workers) from the data. Consequently, we have proposed a new Cost Model to fit the safety of large language models.
> > > > > >
> > > > > > ---
> > > > > >
> > > > > > **Part 2: The Necessity of the Cost-Reward Formulation of CMDP**
> > > > > >
> > > > > > We do not deny that a simple single Reward Model can improve the performance of LLMs. However, **for more efficient simultaneous enhancement of the helpfulness and harmlessness, modeling as a CMDP may be a better choice**. Here are our reasons:
> > > > > >
> > > > > > - **The necessity of modeling human values with multiple models.** Human values are multidimensional, and the relationships between different dimensions (such as helpfulness, harmlessness, honesty, etc. [1]) are complex (e.g., inherent conflicts between helpfulness and harmlessness [2]). Therefore, using **a single Reward Model** has its drawbacks: firstly, it **requires a stronger base model** with greater fitting capacity and more data during the training; secondly, a single Reward Model **provides less information** during the training of the Actor Model. Hence, using multiple models to represent human values is becoming a cutting-edge research topic [3,4]. In our work, we focus on the dimensions of helpfulness and harmlessness in human values, and due to the inherent tension between the two, it is necessary to decouple them into two parts.
> > > > > >
> > > > > > - **The necessity of modeling harmlessness as a constraint (Cost + Threshold)**. Harmlessness needs to be modeled as a constraint for two reasons: first, it should be **satisfied as a priority**; second, their safety **should not be overly optimized when LLMs are sufficiently safe**, as this would drastically reduce helpfulness [2]. Under the CMDP formulation, once our trained large language models reach the safety threshold, Safe RLHF tends to maintain sufficient harmlessness rather than over-optimizing it.
> > > > > >
> > > > > > References:
> > > > > >
> > > > > > [1] Askell, Amanda, et al. "A general language assistant as a laboratory for alignment." (2021).
> > > > > >
> > > > > > [2] Bai, Yuntao, et al. "Training a helpful and harmless assistant with reinforcement learning from human feedback." (2022).
> > > > > >
> > > > > > [3] Bakker, Michiel, et al. "Fine-tuning language models to find agreement among humans with diverse preferences." (2022)
> > > > > >
> > > > > > [4] Yuan, Zheng, et al. "Rrhf: Rank responses to align language models with human feedback without tears." (2023).
> > > > > >
> > > > > > ---
> > > > > >
> > > > > > Finally, we sincerely appreciate your willingness to continue the discussion about our work. Your valuable feedback has been helpful to our paper. If you have any further questions, we would be very pleased to engage in further discussion. Thank you once again for your time and attention.

---

### Official Review · Reviewer_murU · 2023-10-31

**Soundness:** 4 excellent
**Presentation:** 4 excellent
**Contribution:** 4 excellent
**Rating:** 8
**Confidence:** 3

**Summary:**

The paper proposes Safe Reinforcement Learning from Human Feedback (Safe RLHF), a novel algorithm for human value alignment in large language models (LLMs). Safe RLHF decouples human preferences regarding helpfulness and harmlessness, allowing separate training of reward and cost models. The safety concern of LLMs is formalized as an optimization task, and the balance between the two objectives is dynamically adjusted during fine-tuning. Through three rounds of fine-tuning using Safe RLHF, the paper demonstrates improved performance and harm mitigation compared to existing value-aligned algorithms.

**Strengths:**

1. Separation of rewards and costs is an excellent idea that probably resolves the optimization contradiction in RLHF of LLM.
2. The paper provides concrete experimental results demonstrating the effectiveness of Safe RLHF in enhancing model performance and reducing harmful responses.

**Weaknesses:**

Minor suggestions in Questions.

**Questions:**

1. Fig2.(a) the symbols of axis indexes are absent.
2. Could you add the scatter point of Beaver-v2 and Beaver-v3 in Fig.6(a) and (b)?

---

> ### Author Response · Authors · 2023-11-16
> **Official Reply to Reviewer murU**
>
> We greatly appreciate your valuable feedback and are encouraged by your recognition of the Safe RLHF framework as an excellent idea that probably resolves the optimization contradiction in the RLHF of LLM. It is gratifying to know that our efforts and dedication are acknowledged. Should you have any additional questions or suggestions, we warmly invite you to share your insights with us.
>
> ---
>
> > **Q1:** Fig2.(a) the symbols of axis indexes are absent.
>
> **Reply to Q1:** We are very grateful to the reviewer for their comments on Figure 2(a). Indeed, we acknowledge that this figure lacks some necessary symbols and explanations, which we will add both here and in the newly submitted PDF version of the paper.
>
> **The red dashed line in Figure 2 represents the ambiguous boundary between safety and unsafety in human values.** By proposing a new training method for a preference model (cost model), we fit this ambiguous boundary while maintaining the original properties of the Bradley-Terry model. The introduction of this cost model is fundamental for integrating RLHF with Safe RL algorithms and is one of our main contributions. Specifically, building on the traditional preference model, we introduce a comparison loss between responses and a virtual response $y_0$ near the safety threshold, setting the cost of $y_0$ to zero (Equation 4).
>
> ---
>
> > **Q2:** Could you add the scatter point of Beaver-v2 and Beaver-v3 in Fig.6(a) and (b)?
>
> **Reply to Q2:** As illustrated in the following figure, we added scatter plots for Beaver-v2 and Beaver-v3 on the basis of Fig.6(a) and Fig.6(b):
>
> https://anonymous.4open.science/r/Rebuttal-ICLR24/images/ablation.png
>
> However, it is important to note that Beaver-v2 and Beaver-v3 were initialized using Beaver-v1 and Beaver-v2 respectively. Given that these plots employ the win rate against Alpaca-7B as the axis, a direct comparison with other baselines in the figure is unfair. This is the primary reason they were omitted in the initial version of the paper. This is also why we chose to use the Elo score as the evaluation metric for Figure 5.

---

### Official Review · Reviewer_NDaV · 2023-11-07

**Soundness:** 3 good
**Presentation:** 3 good
**Contribution:** 3 good
**Rating:** 8
**Confidence:** 4

**Summary:**

The authors propose safe rlhf, a framework to decouple helpfulness and harmfulness of RLHF model responses. They employ a dynamic λ-trade-off to dual helpfulness and harmlessness objectives. They demonstrate that this approach also results in better inter-rater agreement thereby generating cleaner annotations for RLHF training. They define a cost model for harmlessness and pose the problem as a constrained optimization problem where they employ Lagrangian method to solve the same. Through three rounds of such safe RLHF iterations, they are able to demonstrate at 7B model scales that they are able to mitigate harmful responses while also improving model performance compared to standard value alignment algorithms.

**Strengths:**

With safety being an important aspect in LLMs, this paper tackles an important question -- how to do value alignment under both the safety and usefulness axes. The paper is well written and explains the methodology involved clearly. Even if the techniques to accommodate safety costs into RLHF are simple and straightforward, the paper does a good job in explaining the motivation behind the choices and conducts careful ablations to demonstrate the motivations behind these choices. The evaluation methodology is also robust and the paper carefully evaluates the design choices. The paper also presents a clean framework to decouple different human values safety being one of them from the overall utility of the responses and thus can be extended to other desiderata easily.

**Weaknesses:**

One thing that I feel the paper could do a better job of is to incorporate more safe RLHF baselines. For example, Constitutional AI [1] tackles a very similar problem balancing helpfulness and harmlessness. The only couple of ablations that I can see are of fixed lambda (reward shaping approach) and the approach used in Sparrow. I would have loved to see one or two more safe RLHF approaches that do not need to tow the lines of conventional RLHF exactly.

The improvement achieved in the RLHF stages over the base models is often a function of the SFT stage in between. Aspects of safety can also be incorporated in the SFT data and the paper uses Alpaca 7B off the shelf. If a fine-tuning stage could be done on the responses collected as part of the safety data (or special SFT data can be collected in this regard) and can be introduced as a step in between, then the gains of RLHF with the cost/preference models could be more clearly earmarked compared to the simpler SFT stages. This will help us understand how much value we get by framing this problem during the RL stage vs SFT stage vs both. The focus on the SFT aspect is one thing that find missing in this paper.


1. Constitutional AI: Harmlessness from AI Feedback - Bai et al, Dec 2022.

**Questions:**

1. I think you can do a slightly better job of rewriting the cost model explanation in Section 3.2. Particularly it was a bit time consuming to understand the motivations behind the loss formulation that had both the BT terms and also classification-like term. It would be nice to revisit this explanation in the final draft.

2. More of a suggestion : I would recommend moving important things like Related Work to the main section of the paper instead of leaving them in Appendix. It would be important to position the work in the context of other relevant works. The page limitation could be adjusted by shortening and reformatting other sections or moving some of those to appendix.

3. Were other safe RL baselines considered apart from Sparrow and reward shaping ?

4. Is there any reason why stronger SFT models were not trained considering the fact that you gathered some safety related data before doing RLHF ?

5. I am not sure whether there are clear explanations of how model selection was done both in the RM/Cost model training stage and RLHF training stage. Can you clarify ? Often one finds that model selection plays a much bigger role in final performance than even the training algorithm. Considering you do multiple RLHF iterations, I wanted to understand this aspect clearly.

6. Again a minor suggestion : Some of the labels and text in the figures were hard to read at 100% resolution. Kindly resize them for appropriate reading.

---

> ### Author Response · Authors · 2023-11-16
> **Official Reply to Reviewer NDaV (1/2)**
>
> > **W1:** One thing that I feel the paper could do a better job of is to incorporate more safe RLHF baselines. For example, Constitutional AI tackles a very similar problem balancing helpfulness and harmlessness. The only couple of ablations that I can see are of fixed lambda (reward shaping approach) and the approach used in Sparrow. I would have loved to see one or two more safe RLHF approaches that do not need to tow the lines of conventional RLHF exactly.
>
> > **W2:** The improvement achieved in the RLHF stages over the base models is often a function of the SFT stage in between. Aspects of safety can also be incorporated in the SFT data and the paper uses Alpaca 7B off the shelf. If a fine-tuning stage could be done on the responses collected as part of the safety data (or special SFT data can be collected in this regard) and can be introduced as a step in between, then the gains of RLHF with the cost/preference models could be more clearly earmarked compared to the simpler SFT stages. This will help us understand how much value we get by framing this problem during the RL stage vs SFT stage vs both. The focus on the SFT aspect is one thing that find missing in this paper.
>
> > **Q3:** Were other safe RL baselines considered apart from Sparrow and reward shaping ?
>
> **Reply to W1W2Q3:** Thank you for your constructive feedback regarding our paper. We appreciate the opportunity to enhance our work based on your insightful suggestions. In response to your comments, we have conducted additional experiments with **Constitutional AI** and **Safety SFT** baselines to address both of the weaknesses you identified. Below is a summary of our response and findings:
>
> Win rates for three different methods compared to Aplaca-7B rated by GPT-4:
> | | Safe RLHF (ours; Beaver-v1) | Constitutional AI | Safety SFT |
> | :---: | :---: | :---: | :---: |
> | Helpfulness Win Rate | 71.8% | 40.2% | 53.6% |
> | Harmlessness Win Rate | 73.3% | 69.7% | 53.6% |
>
> Our experiments reveal interesting insights:
>
> **Helpfulness and Harmlessness Balance:** Neither Constitutional AI (40.2% helpfulness, 69.7% harmlessness) nor Safety SFT (53.6% for both) matched the performance of our Safe RLHF (Beaver-v1) approach, which achieved a helpfulness win rate of 71.8% and harmlessness win rate of 73.3%. This underscores **the challenge in balancing helpfulness and harmlessness** in large language models and highlights the efficacy of our approach.
> Impact of SFT on RLHF: Addressing your second concern, our findings suggest that while the SFT stage contributes to model safety, the gains from the RLHF stage, especially with advanced reward models like Beaver-v1, are more pronounced. **This distinction is crucial in understanding the relative impacts of SFT and RLHF stages on model performance.**
> Our additional experiments demonstrate that while incorporating safety in the SFT stage is beneficial, **the RLHF stage offers a more substantial improvement in balancing helpfulness and harmlessness**. The inclusion of new baselines provides a comprehensive view of the current landscape of our Safe RLHF method and validates the effectiveness of our approach.
>
> We believe these enhancements address your concerns and significantly strengthen our paper's contributions to the field of large language models.
>
> **Implementation details of Constitutional AI and Safety SFT:**
> - **Constitutional AI:** We follow the Constitutional AI paper to revise the raw responses based on constitutional critiques. The Constitutional AI method requires a large model with the capability to follow complex instructions to generate critiques for the responses and then revise the responses. We use the GPT-3.5 model to revise the responses generated by Alpaca-7B using the same constitutional prompts from the Constitutional AI paper. We let the reward model tend to prefer the revised response over the original response. And apply the standard RLHF pipeline over the Alpaca-7B model.
>
> - **Safety SFT:** For the Safety SFT method, we first filter out preference pairs in our preference dataset if both responses are labeled unsafe. Then we finetune the Alpaca-7B model to fit the safer response in the preference dataset.

---

> > ### Author Response · Authors · 2023-11-16
> > **Official Reply to Reviewer NDaV (2/2)**
> >
> > > **Q1:** I think you can do a slightly better job of rewriting the cost model explanation in Section 3.2. Particularly it was a bit time consuming to understand the motivations behind the loss formulation that had both the BT terms and also classification-like term. It would be nice to revisit this explanation in the final draft.
> >
> > > **Q2:** More of a suggestion: I would recommend moving important things like Related Work to the main section of the paper instead of leaving them in Appendix. It would be important to position the work in the context of other relevant works. The page limitation could be adjusted by shortening and reformatting other sections or moving some of those to appendix.
> >
> > > **Q6:** Again a minor suggestion : Some of the labels and text in the figures were hard to read at 100% resolution. Kindly resize them for appropriate reading.
> >
> > **Reply to Q1Q2Q6:** We are very grateful to the reviewer for their valuable feedback on the expression (Q1), structure (Q2), and images (Q6) in our paper. We will actively incorporate these improvements in the newly submitted version of our paper (with the modified sections marked in **orange**).
> >
> > ---
> >
> > > **Q4:** Is there any reason why stronger SFT models were not trained considering the fact that you gathered some safety related data before doing RLHF ?
> >
> > **Reply to Q4:** In Section 5.1 Initial SFT Model, we detailed the motivation behind our choice of the initial SFT model in our experiments. The primary reasons for not training stronger SFT models are as follows:
> >
> > - The Alpaca team has made their data, training code, and final model **open source**, which greatly facilitates **replication** by the academic community.
> > - In fact, we **lack the expert data required to train a stronger SFT model**, as the available safety-related data is insufficient.
> > - By using Alpaca-7B as the starting model and applying three rounds of Safe RLHF fine-tuning, we can observe **the effects of Safe RLHF under different levels** of helpfulness and harmlessness.
> >
> > ---
> >
> > > **Q5:** I am not sure whether there are clear explanations of how model selection was done both in the RM/Cost model training stage and RLHF training stage. Can you clarify ? Often one finds that model selection plays a much bigger role in final performance than even the training algorithm. Considering you do multiple RLHF iterations, I wanted to understand this aspect clearly.
> >
> > **Reply to Q5:** Model selection is common in RLHF to ensure **correctness** at every step.
> > To ensure **fairness**, we applied this engineering trick to all types of RLHF algorithms used in our experiments, including Safe RLHF, RLHF (PPO), Reward Shaping, and ablation experiments. In fact, our approach of model selection enhances the fairness of comparisons as it mitigates randomness and ensures each algorithm operates correctly. Here, we will share our approach in detail.
> >
> > **For both the reward model and cost model, the selection of model primarily aims to achieve higher prediction accuracy on test set.** For different parameter training outcomes, we evaluate their predictive accuracy on a reserved test set and select the one with the highest accuracy for the next step. Typically, an accuracy above 75% is considered acceptable by us. With a fixed dataset, the impact of different hyper-parameters on the reward model and cost model is not significant. Therefore, we do not perform model selection repeatedly many times at this stage. For the best hyperparameters, please refer to Appendix G.1.
> >
> > **For the RL phase, model selection primarily aims to prevent over-optimization [1].** Since the reward model and cost model are learned from human preference data, their ability to correctly predict has a range. Continuously training with the same reward and cost models can easily lead to the phenomenon of reward hacking. Therefore, the model selection during the RL phase mainly involves evaluating models at different training steps within the same training cycle to identify the point where the model is sufficiently trained but not over-optimized. Existing evaluations for alignment rely on GPT-4 and human evaluators, and due to their high financial and time cost, we opt for a rapid model selection process that involves human evaluation on a small test set combined with a trained unified score model  (as mentioned in Section 5.2.1 Model-based Evaluations). Only models deemed sufficiently good are then tested with the entire test set using GPT-4 and human evaluations.
> >
> > Also, due to the issue of over-optimization, it is necessary to continually collect data from new models, train new preference models, and perform multiple RLHF iterations.
> >
> > **References:**
> >
> > [1] Gao, Leo, John Schulman, and Jacob Hilton. "Scaling laws for reward model overoptimization." International Conference on Machine Learning. PMLR, 2023.

---

> > > ### Author Response · Authors · 2023-11-20
> > > **Hope to Get Your Reply**
> > >
> > > Dear Reviewer NDaV,
> > >
> > > As the deadline is nearing, we wanted to gently follow up on our recent submission. Your feedback is highly valuable to us, and we would appreciate any updates or further guidance you might have regarding our revisions and responses.
> > >
> > > Thank you for your time and consideration.

---

> ### Comment · Reviewer_NDaV · 2023-11-20
> **Acknowledgement of rebuttal**
>
> Thanks I have read your rebuttal. I am happy with the additional baselines and the authors addressing other questions. Increasing my score.

---

> > ### Author Response · Authors · 2023-11-21
> > **Gratitude to Reviewer NDaV for Recognition**
> >
> > We sincerely appreciate your recognition of our work and the increase in our score. It is an honor for us to address your concerns, and your invaluable insights will be integrated into the final revised version.

---

### Author Response · Authors · 2023-11-16
**General Comments to All Reviewers**

We thank the reviewers (Reviewer NDaV, Reviewer murU, Reviewer xNog, Reviewer NMqi) for their valuable feedback.

We are encouraged that the reviewers found that **our Safe RLHF framework is an excellent and innovative idea** (Reviewer murU, Reviewer NMqi) **which contributes to an important aspect in the domain of LLM safety** (Reviewer NDaV, Reviewer xNog, Reviewer NMqi); **our experiments are robust, comprehensive, and convincing** (Reviewer NDaV, Reviewer murU); and our paper is **well-written and clearly explains the involved methodology** (Reviewer NDaV, Reviewer xNog); **the release of all our data and code is a valuable contribution to research community** (Reviewer NMqi).

We addressed all the reviewer comments below and will incorporate them into the revision. The revised version primarily includes the following significant updates, with the modified sections marked in orange:

1. We increased **the font size** of the images (Reviewers NDaV and NMqi).
2. In the Introduction, we have **more clearly outlined our contributions** (Reviewers xNog and NMqi).
3. We **moved the Related Work section from the appendix to the main text** and provided a more detailed discussion of the challenges in the field of LLM safety (Reviewer NDaV).
4. We **refined the section on the Cost Model** to make it easier to understand (Reviewer NDaV).
5. We **added some missing descriptions in Figures 2 and 4** (Reviewer murU).
6. In Appendix B, we included **ablation experiments on modeling harmlessness using the traditional preference modeling approach** (Reviewer xNog).
7. In Appendix G.2, we described **the motivation and approach for our model selection** (Reviewer NDaV).
8. In Appendix G.3, we added details about **the runtime environment and the costs** involved in our experiments (Reviewer xNog).


If our rebuttal addresses the concerns, we earnestly and kindly ask the reviewers to consider raising the rating and supporting us for acceptance.

---

### Public Comment · ~Han_Zhang3 · 2023-12-10
**How to get the Eq.(29) ?**

I read code/safe_rlhf/algorithms/ppo_lag/trainer.py,  the line 312 is
```
lambda_loss = -(episode_costs - self.threshold) * self.log_lambda.exp()
```
It seems that Eq.(29) is from the line 312:

$\ln \lambda_{k+1} = \ln \lambda_{k} + \alpha \cdot \lambda_{k} \cdot \mathcal{J_C}(\theta_k)$

However, this method of differentiation does not appear to originate from the Lagrange multiplier approach. Does the author provide any explanation for this?

---

> ### Public Comment · ~Weixi_Ning1 · 2023-12-10
> **A Friendly Reminder**
>
> Disclaimer: I am neither an author nor in anyway related to this paper.
>
> Hi Han Zhang, I want to kindly remind you that the rebuttal period has now passed, and the authors are no longer able to provide responses.
>
> It is important to highlight that Equation (29) is consistent with the Lagrange method and its implementation in the code. The authors have applied a common engineering trick. Since $\lambda \geq 0$, they set $\lambda_k \doteq e^{\eta_k}$. By using $\eta$ as the actual update parameter, they ensure $\lambda \geq 0$. Therefore, according to Equation (12):
>
> $$
> \eta_{k+1} = \eta_k + \alpha\frac{\partial}{\partial \eta} e^\eta\mathcal J(\theta_k)\Big |_{\eta_k}
> = \eta_k + \alpha \cdot \mathcal J(\theta_k) e^{\eta_k}
> $$
>
> Substituting in $\eta_k = \ln \lambda_k$, the equation becomes:
>
> $$
> \ln \lambda_{k+1} = \ln \lambda_k + \alpha \cdot \lambda_k \cdot \mathcal J_C(\theta_k)
> $$

---

> > ### Public Comment · ~Han_Zhang3 · 2023-12-11
> > **Thanks for response**
> >
> > Thank you for your response; I have understood the issue.

---

### Meta-Review · Area_Chair_72k3 · 2023-11-30

**Metareview:**

**Summary**: This paper proposes a method (Safe RLHF) for finetuning LLMs by collecting separate labels for "helpfulness" and "harmlessness," which are then maximized in a Lagrangian framework. Experiments apply this approach to Alpaca-7B, showing that the resulting model is better according to human evaluators.

**Strengths**:
Reviewers appreciated both the importance of the problem and the proposed approach. They complimented the robust evaluation, clear motivation, and strong empirical results.

**Weaknesses**:
The main reviewer concern seemed to be that, seen as a safe RL method, the method is fairly mundane (the main significance is its application to LLMs). Reviewers had questions about why the constraint couldn't be baked into the reward function, and had suggestions for additional baselines. The authors did a great job addressing both these points during the rebuttal.

**Justification For Why Not Higher Score:**

The idea is rather straightforward (Lagrangian duality).

**Justification For Why Not Lower Score:**

Reviewers seemed to generally enjoy this paper and found it's contributions useful, significant, and not too complex.

---

### Decision · Program_Chairs · 2024-01-16

Accept (spotlight)